# Membrane properties specialize mammalian inner hair cells for frequency or intensity encoding

Stuart L Johnson*

Department of Biomedical Science, University of Sheffield, Sheffield, United Kingdom

**Abstract** The auditory pathway faithfully encodes and relays auditory information to the brain with remarkable speed and precision. The inner hair cells (IHCs) are the primary sensory receptors adapted for rapid auditory signaling, but they are not thought to be intrinsically tuned to encode particular sound frequencies. Here I found that under experimental conditions mimicking those in vivo, mammalian IHCs are intrinsically specialized. Low-frequency gerbil IHCs (~0.3 kHz) have significantly more depolarized resting membrane potentials, faster kinetics, and shorter membrane time constants than high-frequency cells (~30 kHz). The faster kinetics of low-frequency IHCs allow them to follow the phasic component of sound (frequency-following), which is not required for high-frequency cells that are instead optimally configured to encode sustained, graded responses (intensity-following). The intrinsic membrane filtering of IHCs ensures accurate encoding of the phasic or sustained components of the cell's in vivo receptor potential, crucial for sound localization and ultimately survival.

*For correspondence: s.johnson@ sheffield.ac.uk

**Competing interests:** The author declares that no competing interests exist.

## Introduction

Sound pressure waves are used by many animal species to gain sensory input from their environment that is vital for their survival and communication. The sensory receptors of the auditory system, the hair cells, are responsible for the transduction of sound waves into electrical signals that travel along the auditory pathway. The mechanosensory apparatus of hair cells is the stereociliary hair bundle that protrudes from their apical surface in the form of a staircase-like structure (*Schwander et al., 2010*). Mechano-sensitive ion channels are positioned near the tip of the shorter rows of stereocilia (*Beurg et al., 2009*), the opening of which initiates a receptor potential that is shaped by voltage and ion-gated ion channels in the hair cell's basolateral membrane. The receptor potential activates the $Ca^{2+}$ channels in the basal pole of the cell that trigger the release of neurotransmitter onto afferent terminals (*Glowatzki and Fuchs, 2002*), so that sound information is conveyed to the higher auditory brain areas.

Auditory hair cells in lower vertebrates such as the turtle and bullfrog (*Fettiplace and Fuchs, 1999*), where the hearing sensitivity spans a generally low-frequency range (from around 100 Hz to 1 kHz), show intrinsic membrane tuning such that the membrane potential oscillates at the cells' characteristic frequency (CF). In these animals, the interplay between the $Ca^{2+}$ and $Ca^{2+}$-activated $K^+$ channels, which are differentially expressed along the auditory organ, are thought to be the main mechanism determining hair cell frequency selectivity (*Hudspeth and Lewis 1988*; *Wu et al., 1995*). The mammalian auditory system has evolved to be able to encode a much wider dynamic range of sound frequencies and intensities. In order to do this, mammals have developed an elaborate auditory organ where much of the frequency tuning is carried out by the mechanics of the basilar membrane (*Robles and Ruggero, 2001*) and electromotility of the outer hair cells (OHCs)

**eLife digest** Many animals' survival depends on them accurately and quickly identifying sounds in their environment. In animals with backbones, cells with hair-like projections (called hair cells) inside the ear convert information collected from sound waves into electrical signals. These signals are then transmitted to the brain, which processes the information further.

Animals like bullfrogs are adapted to hearing low frequency sounds, like their own mating calls. These frog's hair cells are individually tuned so that they can capture sounds in this low frequency range. Mammals, on the other hand, have evolved to hear a much wider range of sounds from loud and low frequency sounds, such as thunder, to soft and high frequency sounds, like the cries of their young. In mammals, the part of inner ear involved in hearing (called the cochlea) has an elaborate spiral-like shape. The structure of the cochlea results in different frequencies of sound being transformed by the hair cells into electrical signals at different points around the spiral. Because of this, most researchers didn't think that hair cells in mammals were individually tuned like those in bullfrogs.

Now, Stuart Johnson demonstrates that hair cells in different parts of the gerbil's cochlea are specialized for encoding sounds of specific frequencies. In conditions that mimic the environment inside the ear, a very precise jet of fluid was used to stimulate single hair cells in a similar way to a sound wave. The experiments then compared how hair cells from the upper and lower parts of the cochlea's spiral responded. Johnson found that hair cells from the upper portion of the gerbils' cochlea are specialized to capture low frequency sounds. They have electrical properties that allow them to quickly transmit information to the brain about low frequency sounds. In the lower portion of the cochlea, hair cells are specialized to capture high frequency sounds. That is, their electrical properties make it easier for these hair cells to transmit detailed information to the brain about the volume of high frequency sounds. Together, these findings help explain how these animals are able to localize sounds, which requires capturing both the timing and intensity of different types of sounds.

(*Brownell et al., 1985*; *Ashmore, 1987*) driven by the motor protein prestin (*Liberman et al., 2002*; *Dallos et al., 2006*). These morphological and functional differences along the mammalian cochlea produce a tonotopic place-frequency map that is preserved throughout the auditory pathway. Thus the main sensory receptors, the inner hair cells (IHCs), are believed not to show intrinsic tuning in mammals (*Marcotti et al., 2003*), although recent studies have shown tonotopic differences in the $Ca^{2+}$ currents (*Johnson and Marcotti, 2008*), the size of $Ca^{2+}$ hotspots at ribbon synapses (*Meyer et al., 2009*), and $Ca^{2+}$ dependence of exocytosis (*Johnson et al., 2008*) along the cochlea. Moreover, the in vivo receptor potentials of low and high-frequency mammalian IHCs differ because of the relative filtering properties of the cells' basolateral membrane. Low-frequency IHCs (up to a few kilohertz) have a predominantly phasic (AC) component that follows the stimulation frequency and is graded in size to the intensity (*Dallos, 1985*). The receptor potentials of high-frequency cells cannot follow the sound stimulus and as such do not have a large phasic component. Instead, they have a predominantly sustained (DC) component that is the average depolarization resulting when there is an asymmetric transduction current and it oscillates too fast for the cell membrane potential to keep up. The DC component is graded in size to sound intensity (*Russell and Sellick, 1978*).

In the present study, I have investigated the biophysical properties of mature gerbil IHCs under near physiological recording conditions in low-frequency (apical turn: ~300 Hz) and high-frequency (basal turn: ~30 kHz) regions of the cochlea. The results show that the size of the resting mechano-transducer (MT) current and the composition of basolateral membrane currents are optimized to confer low- and high-frequency IHCs with different resting membrane potentials ($V_m$) and kinetics in vivo. These biophysical differences allow apical IHCs to follow precisely the phasic component of sound, while basal cells are able to accurately encode the sustained and graded component of high-frequency stimuli, providing direct evidence for the existence of intrinsic filtering in mammalian cochlear IHCs.

## Results

### MT currents in mature gerbil apical and basal coil IHCs

MT currents from apical (~300 Hz) and basal (~30 kHz) coil IHCs of the mature gerbil (P17–P25) were elicited by displacing their hair bundles along their axis of mechanosensitivity with a 50 Hz sinusoidal stimulus using a piezoelectric fluid jet stimulator. The fluid jet produces a uniform deflection of the hair bundle (*Corns et al., 2014*) and for IHCs, which are stimulated in vivo by fluid motion within the cochlear partition (*Fettiplace and Kim, 2014*), it provides a near-physiological stimulation without affecting their resting position. To mimic in vivo conditions as closely as possible, recordings were obtained from IHCs maintained at body temperature and their hair bundles were perfused with an extracellular solution containing an endolymphatic low-$Ca^{2+}$ concentration (40 µM: see Materials and methods). The same low-$Ca^{2+}$ solution was also used inside the fluid jet. A large inward MT current was elicited in both apical (1.1 ± 0.1 nA, n = 5, P20–P25: *Figure 1A*) and basal coil IHCs (1.0 ± 0.2 nA, n = 6, P17–P25: *Figure 1B*) upon moving their bundle in the excitatory direction (i.e., towards the taller stereocilia using positive driver voltages). Apical cells had a significantly larger resting MT current (464 ± 29 pA: *Figure 1A*, P<0.0001), which shut off during inhibitory phases of the stimulus, than that recorded from basal cells (70 ± 17 pA: *Figure 1B*). This difference was also observed with FM1-43, a styryl dye that permeates the open transducer channels and accumulates inside the hair cells (*Gale et al., 2001*; *Furness et al., 2013*). Bath application of FM1-43 onto P18–P22 gerbil cochleae (n = 6) labeled IHCs strongly in the apical coil but only weakly in the base (*Figure 1C and D*, respectively). The opposite was observed in the OHCs, consistent with the fact they have larger resting MT currents in the basal region (*Johnson et al., 2011*).

### Resting potential and voltage responses of IHCs in endolymph-like $Ca^{2+}$

The likely resting potential of mature gerbil IHCs in vivo was determined by performing whole-cell current clamp experiments with endolymphatic low-$Ca^{2+}$ solution bathing the stereociliary hair bundle. Current clamp recordings were performed on mature apical and basal coil gerbil IHCs (P17–P22) in response to current injection steps of up to 1 nA from their resting potential (*Figure 2A–C*). In the presence of 1.3 mM extracellular $Ca^{2+}$, which is normally used when studying basolateral membrane currents, the resting potentials of apical (–68.6 ± 1.3 mV, n = 12) and basal (–71.7 ± 0.6 mV, n = 11) IHCs were very similar (*Figure 2A and D*), and within the same range previously found in mature IHCs from mice (*Marcotti et al., 2004*). The low (40 µM) extracellular $Ca^{2+}$ solution caused apical IHCs to depolarize (–56.1 ± 1.5 mV, n = 12) significantly more than basal IHCs (P<0.0001; –65.8 ± 1.3 mV, n = 11; *Figure 2B and D*). The voltage responses of apical IHCs showed an initial decline and compression for the larger depolarizing current steps, whereas basal were larger, more sustained, and graded to the stimulus, especially in low $Ca^{2+}$ (*Figure 2B and E*). In the presence of the MT channel blocker dihydrostreptomycin (DHS) (100 µM) the voltage responses returned to control levels (40 µM $Ca^{2+}$ + DHS: *Figure 2C and D*), indicating that the shift in the resting membrane potential in low-$Ca^{2+}$ was due to the increased MT channel open probability. These findings demonstrate that the resting membrane potential of apical IHCs is likely to be substantially more depolarized than that of basal cells in vivo, which would speed up their membrane time constant. The time constant of the voltage rise, following 100 pA depolarizing current injections, were significantly more rapid in apical than in basal IHCs in both control 1.3 mM $Ca^{2+}$ (P<0.001; apical: 1.03 ± 0.11 ms, n = 12; basal: 1.89 ± 0.19 ms, n = 11) and low-$Ca^{2+}$ conditions (P<0.0005; apical: 0.44 ± 0.08 ms, n = 12; basal: 1.07 ± 0.13 ms, n = 11) (*Figure 2F*). The different voltage responses between apical and basal IHCs indicates the existence of cellular specializations in the underlying basolateral membrane currents.

### Tonotopic differences in IHC basolateral membrane currents

Basolateral membrane currents were recorded from gerbil IHCs at body temperature in response to 10 mV voltage steps from the holding potential of –64 mV (*Figure 3A and B*). Both apical (n = 26) and basal IHCs (n = 27) showed the currents characteristic of mature IHCs (*Marcotti et al., 2004*; *Kros et al., 1998*), namely, the rapidly activating large conductance (BK), $Ca^{2+}$-activated $K^+$ current ($I_{K,f}$), and the negatively activating delayed rectifier current ($I_{K,n}$: carried by KCNQ4 channels; potassium channel, voltage gated KQT-like subfamily Q, member 4). The overall size

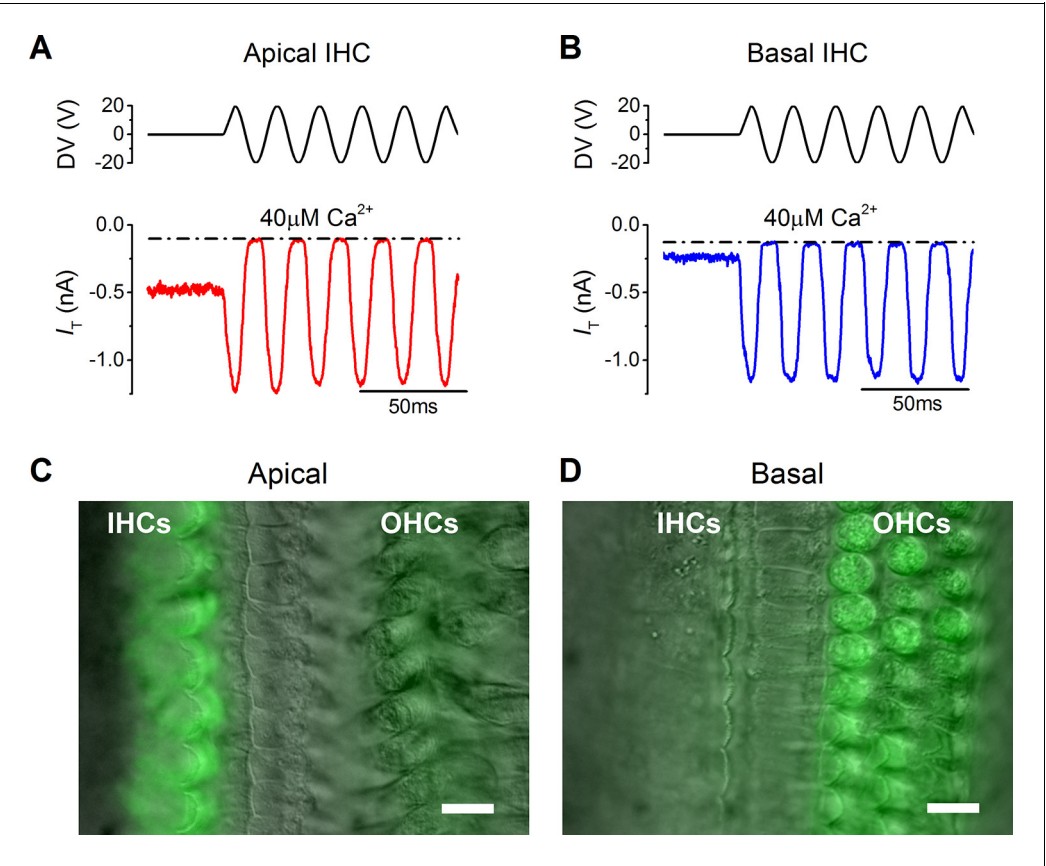

**Figure 1.** The resting mechanotransducer (MT) current in apical and basal inner hair cells (IHCs) from adult gerbils. (A and B) Saturating MT currents recorded from an apical (A) and a basal IHC (B) by applying 50 Hz sinusoidal force stimuli to the hair bundles at the membrane potential of –84 mV. Recordings were carried out in the presence of a low-Ca$^{2+}$ (40 µM) extracellular solution. The driver voltage (DV) signal of up to ± 20 V to the fluid jet is shown above the traces (positive deflections are excitatory). The dashed lines indicate the closure of the transducer channels and disappearance of the resting current during inhibitory bundle displacements. (C and D) FM1-43 fluorescence images of apical (C) and basal (D) gerbil cochlear sections (P18) with the differential interference contrast (DIC) image superimposed. Apical IHCs, and to a much less extent outer hair cells (OHCs), were labeled by the dye whereas the opposite pattern was seen in basal cells. Scale bar represents 10 µm.

of the peak and steady-state inward and outward currents appeared comparable between apical and basal IHCs (*Figure 3C*). The peak current measured at 2 ms gives a reliable and accurate measurement of the size of $I_{K,f}$ in isolation due to its very rapid activation kinetics (*Marcotti et al., 2004*), which was similar between apical and basal gerbil IHCs. The overall slope conductance, measured at around the likely in vivo membrane potential (Ac: –54 mV; Bc: –64 mV: *Figure 2D*) from the steady-state current-voltage (I-V) curves, was found to be significantly larger in apical (56.3 ± 3.0 nS, n = 26, P<0.0001) than in basal IHCs (10.0 ± 0.4 nS, n = 27). Profound differences were observed in the current activation time-course and size of the tail currents between apical and basal IHCs (apical: *Figure 3D, F, and H*; basal: *Figure 3E, G, and H*). All IHCs investigated showed a very negative current activation (*Figure 3H*, inset). Pharmacological analysis revealed a differential expression of K$^+$ currents between apical and basal IHCs (*Figure 4A–E*). The main difference between the cells was the outward delayed-rectifier K$^+$ current that was mainly a linopirdine-insensitive $I_{K,s}$ in basal IHCs (*Figure 3D*) while in apical cells it was predominantly a linopirdine-sensitive current (*Figure 3E*) reminiscent of the outward $I_{K,n}$ found in adult OHCs (*Marcotti and Kros, 1999*). The deactivating tail currents of the linopirdine-sensitive current (*Figure 4F*) indicate that there is more than one linopirdine-sensitive current component. The negatively activating component is larger in basal cells (*Figure 4G*) and is similar in size and voltage-dependence to the $I_{K,n}$ previously shown in apical-coil mouse IHCs

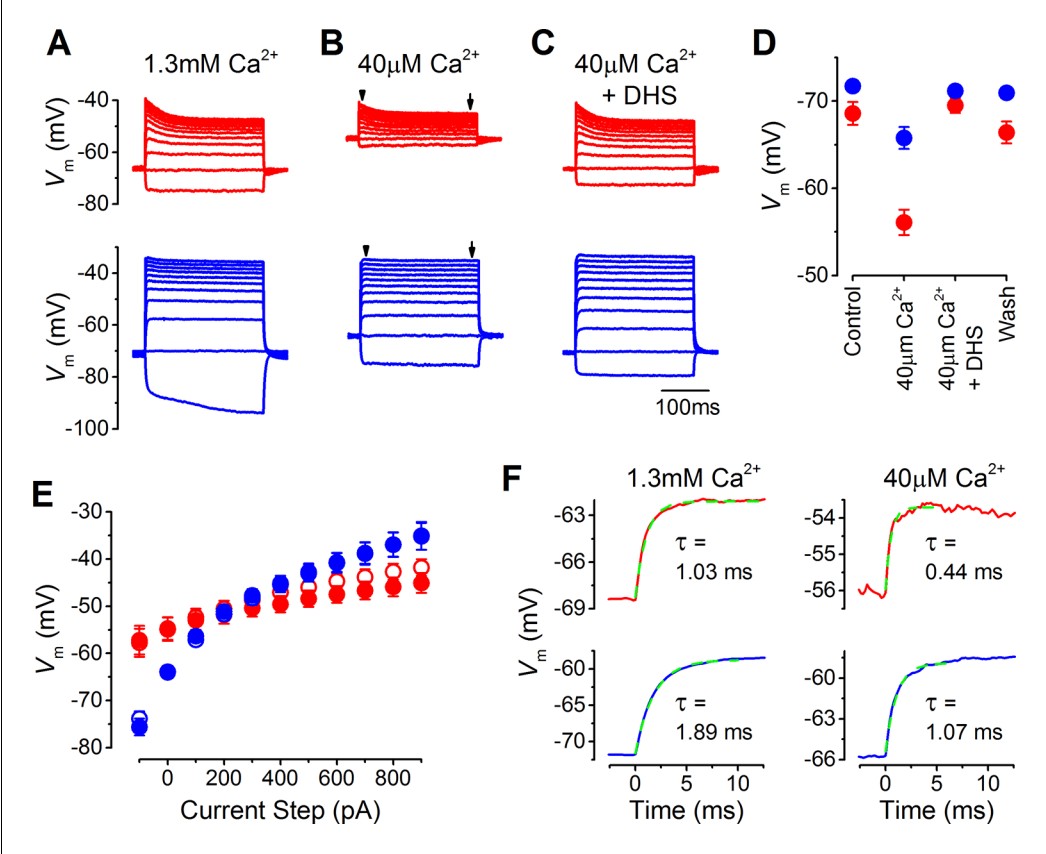

**Figure 2.** The resting membrane potential in apical and basal inner hair cells (IHCs) from adult gerbils. (A–C) Current clamp responses of apical (top panels) and basal (lower panels) IHCs from adult gerbils in response to 100 pA current steps from -100 pA to 900 pA from the cells' resting potential (V_m). From left to right are responses in 1.3 mM extracellular $Ca^{2+}$ (A), low-$Ca^{2+}$ (40 μM) endolymph-like solution (B), and low-$Ca^{2+}$ solution with 100 μM dihydrostreptomycin (DHS) (C). Traces are averages from all IHCs (apical: n = 12; basal: n = 11). (D) Average V_m values measured before the current steps in the different extracellular conditions as in (A–C), including the washout with 1.3 mM $Ca^{2+}$, for apical (red) and basal (blue) IHCs. (E) Average peak (open symbols) and steady-state (closed symbols) V_m from apical and basal IHCs measured at different current injection levels from the current clamp recordings in low-$Ca^{2+}$ in (B). The arrowheads/arrows in (B) indicate where the peak and steady state V_m were measured. Note that the voltage responses of basal cells to a sustained current step, similar to a high-frequency tone, are more clearly graded to the stimulus amplitude than those in apical cells in terms of the dynamic range of voltages covered. (F) Average onset of the initial V_m response to 100 pA current injection in 1.3 mM $Ca^{2+}$ (left) and low-$Ca^{2+}$ (right) for apical IHCs (top panels, n = 12) and basal IHCs (bottom panels, n = 11). The initial rise to peak was fit with a single exponential function. The average time constant (τ), obtained from fitting the individual cells, was (apical IHCs) 1.03 ± 0.11 ms in 1.3 mM $Ca^{2+}$ and 0.44 ± 0.08 ms in low-$Ca^{2+}$; (basal IHCs) 1.89 ± 0.19 ms (1.3 mM $Ca^{2+}$) and 1.07 ± 0.13 (low-$Ca^{2+}$).

that are also comparatively high-frequency receptors (*Marcotti et al., 2003*; *Oliver et al., 2003*). The tonotopic difference in this negatively activating $I_{K,n}$ is consistent with the expression gradient of KCNQ4 channels observed in the rat (*Beisel et al., 2005*). The more positively activating linopirdine-sensitive component was much larger in apical IHCs and could result from the expression of a different KCNQ channel subtype.

Possible differences in the endogenous $Ca^{2+}$ buffering in IHCs along the cochlea (gerbil: *Pack and Slepecky, 1995*) could affect $Ca^{2+}$ loading and lead to differences in the basolateral membrane current properties or MT channel resting open probability. However, recent electrophysiological observations have indicated that the buffering strength in gerbil IHCs along the cochlea is likely to be similar (*Johnson et al., 2008*). In order to verify this observation, I performed recordings from mature gerbil IHCs (P29–P36) under the perforated patch configuration, a condition that prevents the dialysis of the intracellular $Ca^{2+}$ binding proteins with the $Ca^{2+}$ buffer in the pipette solution and therefore preserves any tonotopic gradient that could be present. The basolateral currents recorded using perforated patch (*Figure 5A–C*) were similar to those obtained with whole cell recording (*Figure 3A–C*). The overall current size was similar between apical and basal IHCs (*Figure 5C*), but

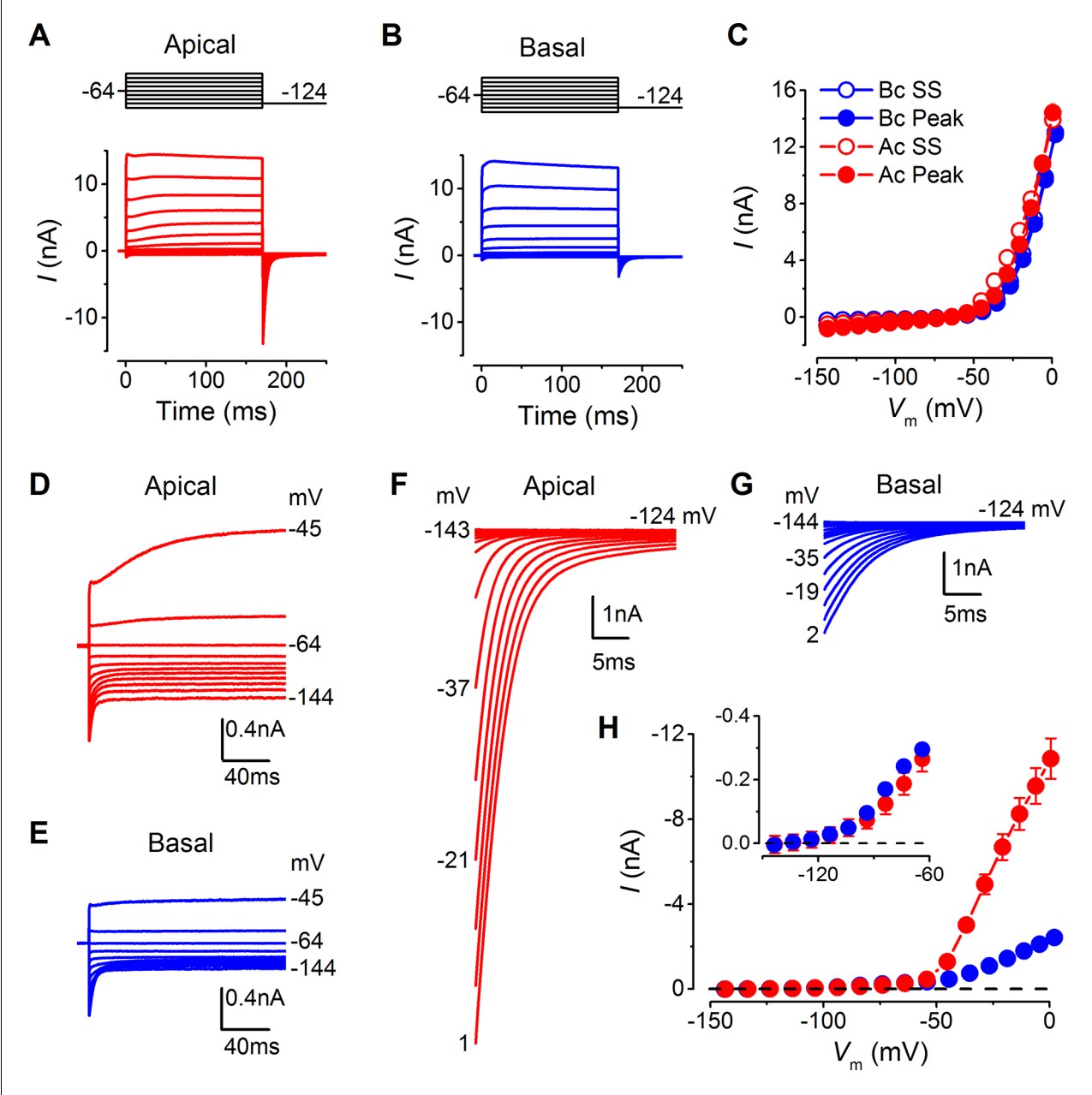

**Figure 3.** Differences in the basolateral membrane currents of apical and basal adult gerbil inner hair cells (IHCs). (A and B) Basolateral membrane currents elicited from apical (A, red) and basal (B, blue) IHCs in response to depolarizing voltage steps in nominal 10 mV increments from −144 mV (holding potential: −64 mV). Tail currents were measured upon returning to −124 mV (protocol shown above the current traces). Traces are averages from 26 apical and 27 basal IHCs. (C) Average current-voltage (I-V) curves for apical and basal IHCs. Peak currents were measured at 2 ms from the onset of the test step and steady-state values were obtained at around 160 ms. (D and E) Average current traces as in (A) and (B), but on an expanded y-scale to show more clearly the time course of the currents at more negative membrane potentials. (F and G) Average tail currents from apical and basal IHCs, respectively, from the different test potentials shown by some of the traces (amplified from [A] and [B]). (H) Average activation curves for apical and basal IHCs made by plotting the instantaneous tail current amplitude against the preceding voltage step. The inset shows the activation of the K⁺ currents on an expanded scale.

the tail currents were much larger in apical cells (*Figure 5A and B*) as shown in *Figure 3*. The size of the resting transducer current, which contributes to the cells holding current in voltage clamp, was measured by perfusing IHCs with low-Ca$^{2+}$ extracellular solution (*Figure 5D and E*). The addition of DHS to the low-Ca$^{2+}$ solution largely prevented the increase in holding current magnitude,

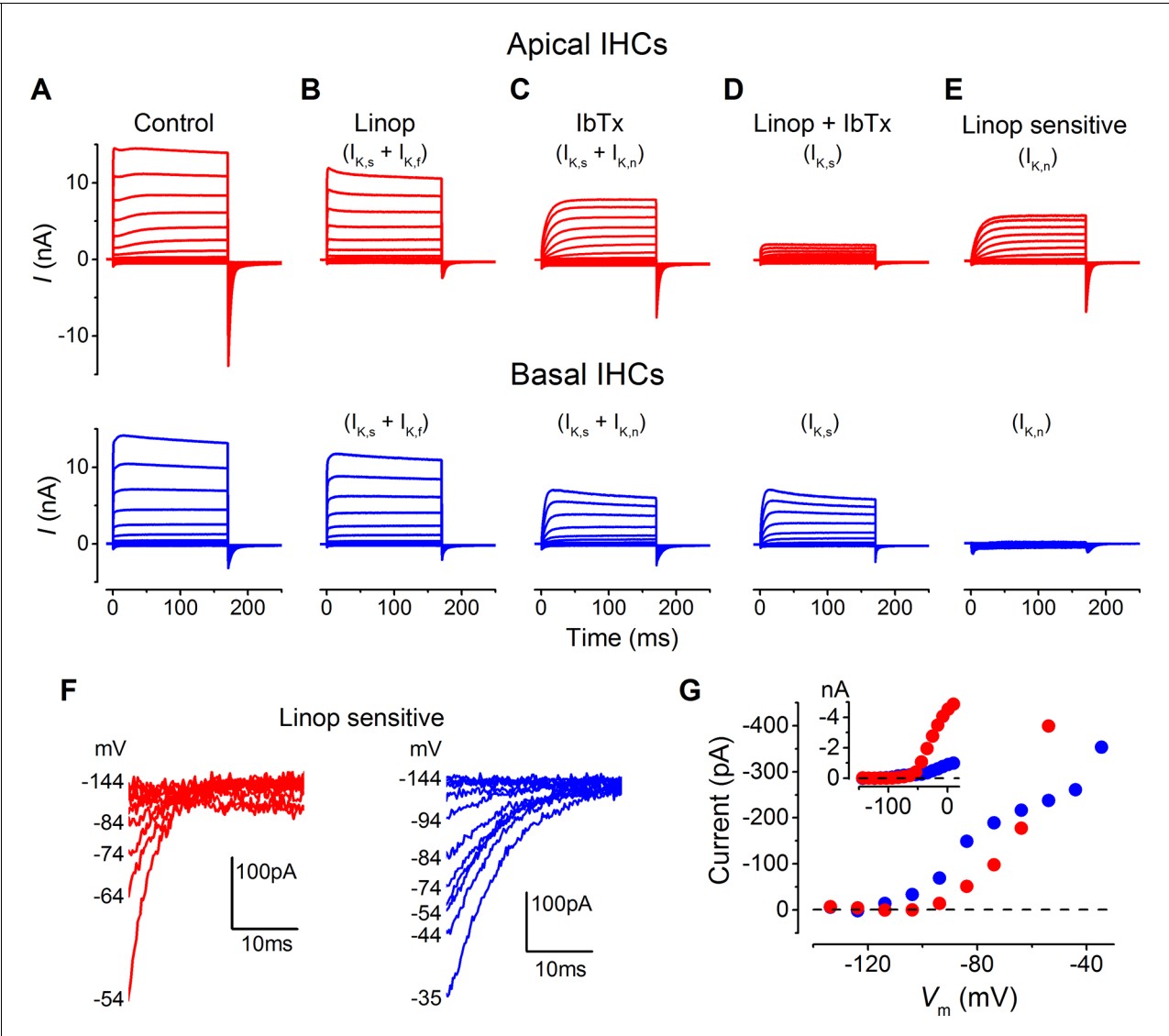

**Figure 4.** Differences in the $K^+$ current components in apical and basal gerbil inner hair cells (IHCs). (A–E) average basolateral currents elicited as in *Figure 3* from apical (red) and basal (blue) IHCs in the presence of different $K^+$ channel blockers to isolate the underlying current components. The control traces (A) are the same as those in *Figure 3A and B*. (B) The perfusion of linopirdine (80 µM) blocked the inward $I_{K,n}$ in both apical and basal IHCs leaving $I_{K,s}$ and $I_{K,f}$ (apical n = 5, basal n = 7). Linopirdine also removed the outward delayed component in apical cells but not in basal cells. (C) The perfusion of iberiotoxin (IbTx; 60 nM) was used to block the rapidly activating $I_{K,f}$ leaving a current composed of $I_{K,s}$ and $I_{K,n}$ (apical n = 4, basal n = 4). (D) The combination of linopirdine and iberiotoxin (Linop + IbTx) was used to isolate $I_{K,s}$ (apical n = 4, basal n = 4), which was small in apical cells but of a similar size to that in C for basal cells. (E) The isolated $I_{K,n}$ (Linop sensitive) was obtained by subtracting the current in the presence of both linopirdine and IbTx (D) from the current obtained in the presence of IbTx (C). (F) Average tail currents at –124 mV from the linopirdine sensitive current in E for apical and basal IHCs. Only the most negative traces are shown to emphasize the negatively activating component of the linopirdine-sensitive current. (G) Average activation curves for the negatively activating component of the linopirdine-sensitive current from the instantaneous tail current amplitudes in F for apical and basal cells. The inset shows the activation curves up to more positive voltages as in *Figure 3H*.

indicating that the transducer channel component was absent. The change in holding current in perforated patch (apical IHCs –764 ± 28 pA, n = 3; basal IHCs –99 ± 21 pA, n = 4) was similar to that obtained with whole cell recordings (apical IHCs –549 ± 81 pA, n = 6; basal IHCs –69 ± 22 pA, n = 4) and also similar to the resting transducer current values obtained from direct MT current measurements in low-$Ca^{2+}$ (*Figure 1A and B*).

These findings show that the biophysics of low-frequency and high-frequency IHCs, including resting MT current size, $K^+$ channel composition, and voltage responses, differ between low-frequency

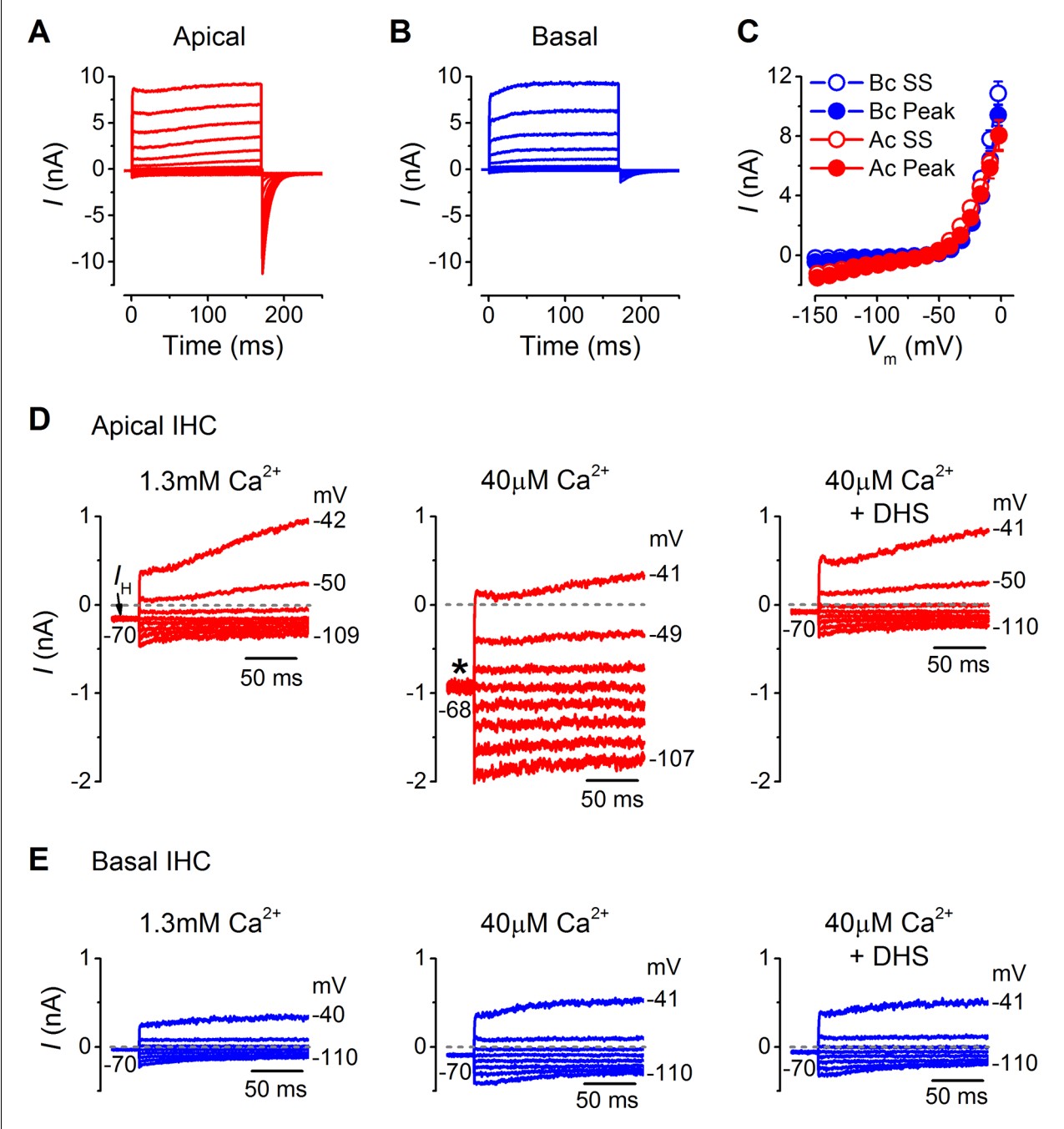

**Figure 5.** Endogenous Ca²⁺ buffering does not affect differences in basolateral membrane currents or the resting mechanotransducer (MT) current in apical and basal adult gerbil inner hair cells (IHCs). (**A** and **B**) Basolateral membrane currents elicited from apical (**A**, red) and basal (**B**, blue) IHCs as in *Figure 3A and B*. (**C**) Average peak and steady-state current-voltage (I-V) curves for apical (n = 3) and basal (n = 4) IHCs. (**D** and **E**) Membrane currents covering a physiological range of IHC membrane potentials elicited from the holding potential of around -70 mV. The holding current is indicated in the first panel as $I_H$. Recordings were made in standard 1.3 mM extracellular Ca²⁺ (left panels), 40 µM Ca²⁺ (middle panels), and 40 µM Ca²⁺ with 100 µM DHS (right panels). Low Ca²⁺ (40 µM) elicited a large inward holding current only in apical IHCs (asterisk) that was blocked by DHS.

and high-frequency IHCs, with apical IHCs having a more depolarized resting $V_m$ and faster membrane time constant in vivo.

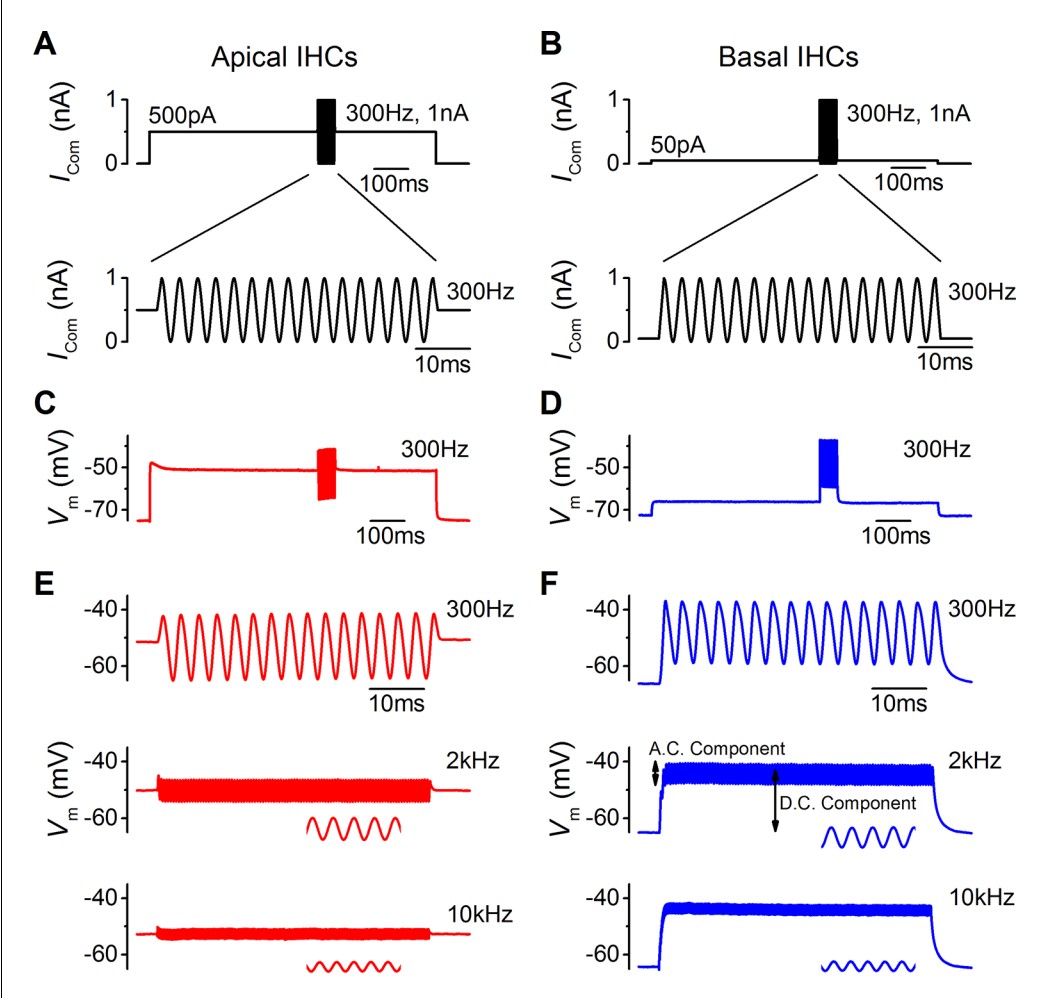

**Figure 6.** Sound-like stimulation of apical and basal adult gerbil inner hair cells (IHCs). (**A** and **B**) Current clamp protocols used to mimic the in vivo sound-induced stimulation of apical (**A**) and basal (**B**) IHCs. A 300 Hz sinewave of 1 nA peak-to-peak amplitude was superimposed on a step current injection (apical: 500 pA; basal: 50 pA), which was used to depolarize lHCs to their predicted in vivo resting potential (see Results). Expanded versions of the sinewave stimulus (top) are shown below. (**C** and **D**) Average voltage responses from apical (n = 7) and basal (n = 6) IHCs in response to the whole stimulus shown in (**A**) and (**B**) (top). (**E** and **F**) Average voltage responses of apical (**E**) and basal (**F**) IHCs to a 300 Hz (top), 2 kHz (middle), and a 10 kHz (bottom) 1 nA sinewave. Only the sinewave portion of the responses is shown and the holding current levels were as in (**A**) and (**B**), respectively. An expanded portion of the 2 kHz and 10 kHz voltage responses is shown below the main trace in order to show some cycles of the sine wave. For the basal response at 2 kHz the predominantly phasic (AC) and predominantly sustained (DC) components are indicated. Apical cells showed only the AC component for 1 nA stimuli.

## Voltage responses of IHCs to sound-like stimulation

The functional implication of the biophysical differences in the resting MT current and voltage responses between low-frequency and high-frequency IHCs (*Figures 1–5*) was investigated by using a sound-like sinusoidal current stimulus applied to IHCs maintained at their predicted in vivo resting potential (*Figure 6*). The sinewave stimulus was superimposed on a sustained holding current, equivalent in value to the resting MT current size measured in endolymphatic low-$Ca^{2+}$ (apical: 500 pA; basal: 50 pA; *Figure 1*).

The sinewave, 1 nA in amplitude and of varying frequency, was used to mimic the transducer current during the sound induced deflection of the hair bundle that would open and close the MT channels from their resting state (*Figure 6A and B*). The averaged voltage responses of apical (n = 7) and basal (n = 6) IHCs to the full protocol, with a 300 Hz sinewave, are shown in *Figure 6C and D*, respectively. Apical cells responded to the different frequencies used (300 Hz, 2 kHz, and 10 kHz

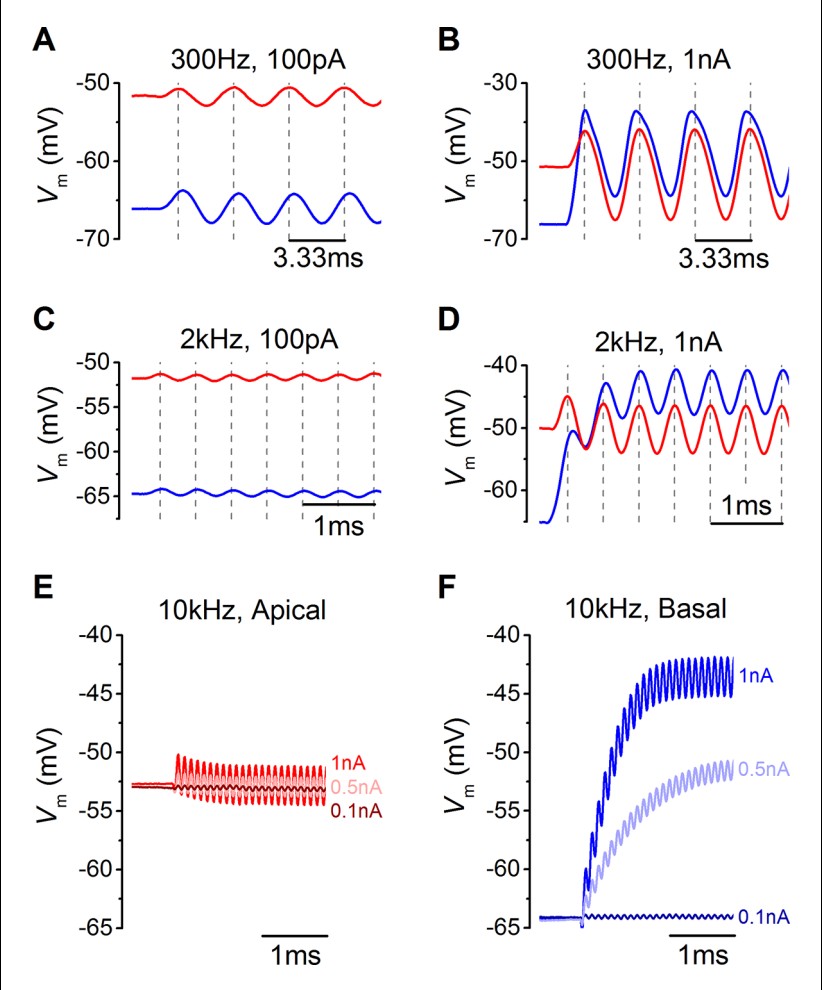

**Figure 7.** Voltage responses of apical and basal inner hair cells (IHCs) to sound-like stimulation of different intensity. (**A**) Voltage responses of apical and basal IHCs to 100 pA, 300 Hz sinewave stimulation. Dashed vertical lines are spaced at 3.33 ms to represent one cycle at 300 Hz with the first being aligned to the initial peak of the apical response. Basal responses show a delay compared to the peaks in apical cells. (**B**) Voltage responses as in (**A**) but using a 1 nA, 300 Hz sinewave (only the initial four cycles are shown). Dashed vertical lines are aligned as in (**A**). Note the leftward leaning of the basal upper peaks following the initial one. (**C**) Responses to 100 pA, 2 kHz stimulation showing the initial few milliseconds. The dashed vertical lines are as in (**A**) but spaced to show cycles at 2 kHz. The responses of apical and basal IHCs are reduced in magnitude compared to those at 300 Hz (**A**), but the delay in the basal peaks was maintained. (**D**) Responses to 1 nA, 2 kHz stimulation with vertical lines as in (**C**). Note that the depolarization builds up in basal responses and that the peaks are delayed compared to those for apical cells. (**E** and **F**) The initial few ms of 10 kHz responses to three different amplitude stimuli, shown next to the traces for apical (**E**, n = 6) and basal cells (**F**, n = 6). While both apical and basal IHCs have a graded phasic (AC) response to the stimulus, basal cells show a pronounced graded (DC) response to the stimulus intensity.

sinewave) with a phasic (AC) component around the resting potential that became progressively smaller with increasing frequency (*Figure 6E*). In basal IHCs the phasic responses to all three frequencies (*Figure 6F*) were superimposed on a sustained shift in membrane potential (DC component). Note that the voltage recordings obtained using sinewave frequencies of 300 Hz (*Figure 6E*, top) and 10 kHz (*Figure 6F*, bottom) are likely to be the closest approximation to in vivo responses for apical and basal IHCs, respectively.

In addition to frequency, IHC responses are also determined by differences in sound intensity, which was mimicked by varying the amplitude of the sinewave (*Figure 7*). Both apical and basal IHCs responded with very small changes in membrane potential when small amplitude sinewaves were

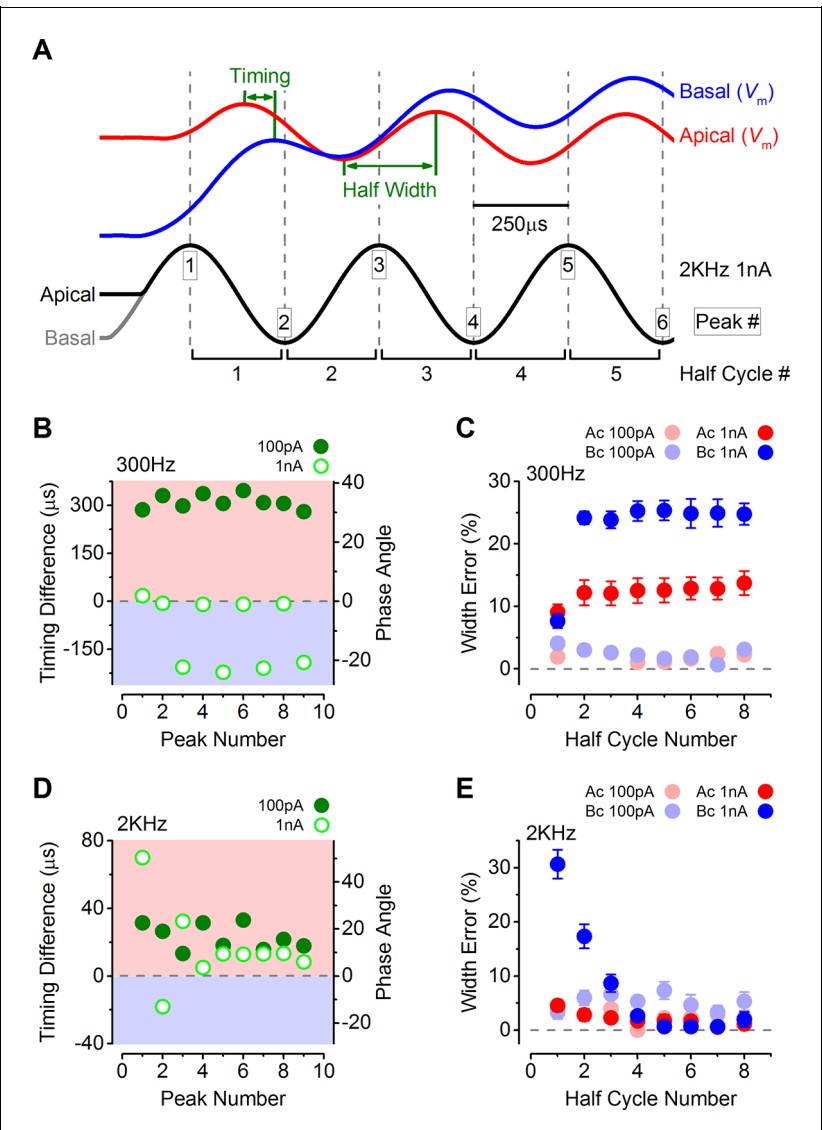

**Figure 8.** The timing of phasic responses to sound-like stimulation is faster and more accurate in apical inner hair cells (IHCs). (**A**) IHC responses to a 1 nA, 2 kHz stimulus showing the relative timing shift between apical and basal cells (above) to the sinewave stimuli (below; apical black and basal grey). Dashed vertical lines delineate each half cycle of the 2 kHz stimulus with the half-cycle number shown. The numbers on the dashed vertical lines indicate the maxima and minima peak number used for subsequent analysis. Note that the responses of apical and basal IHCs are both delayed compared to the stimulus but the corresponding peaks and half cycles can be seen. The difference in timing between apical and basal traces is shown and was measured as the time delay between the occurrences of the equivalent peaks. The half widths were measured as the time between maximal and minimal peaks for each response half cycle. (**B**) The average timing difference between the arrival of the different peaks between apical and basal responses for 300 Hz stimuli (the symbols represent two stimulus amplitudes). Data points were obtained by subtracting the average apical value from that of basal cells. Therefore positive values (light red area) indicate an apical lead whereas negative values (light blue area) indicate a basal lead. The right-hand scale shows the corresponding shift in phase angle for the equivalent timing differences, calculated based on the size of the time difference and the stimulus frequency. (**C**) Average half-cycle width errors for apical and basal responses to 300 Hz stimuli of different amplitudes. Errors were calculated for each cell as the width of each half cycle in milliseconds divided by the actual half-cycle width of the stimulus (1.67 ms for 300 Hz) and converted to percentage. (**D** and **E**) Timing differences and half width errors, respectively, calculated as in (**B**) and (**C**) but for 2 kHz stimuli (half-cycle width of 0.25 ms).

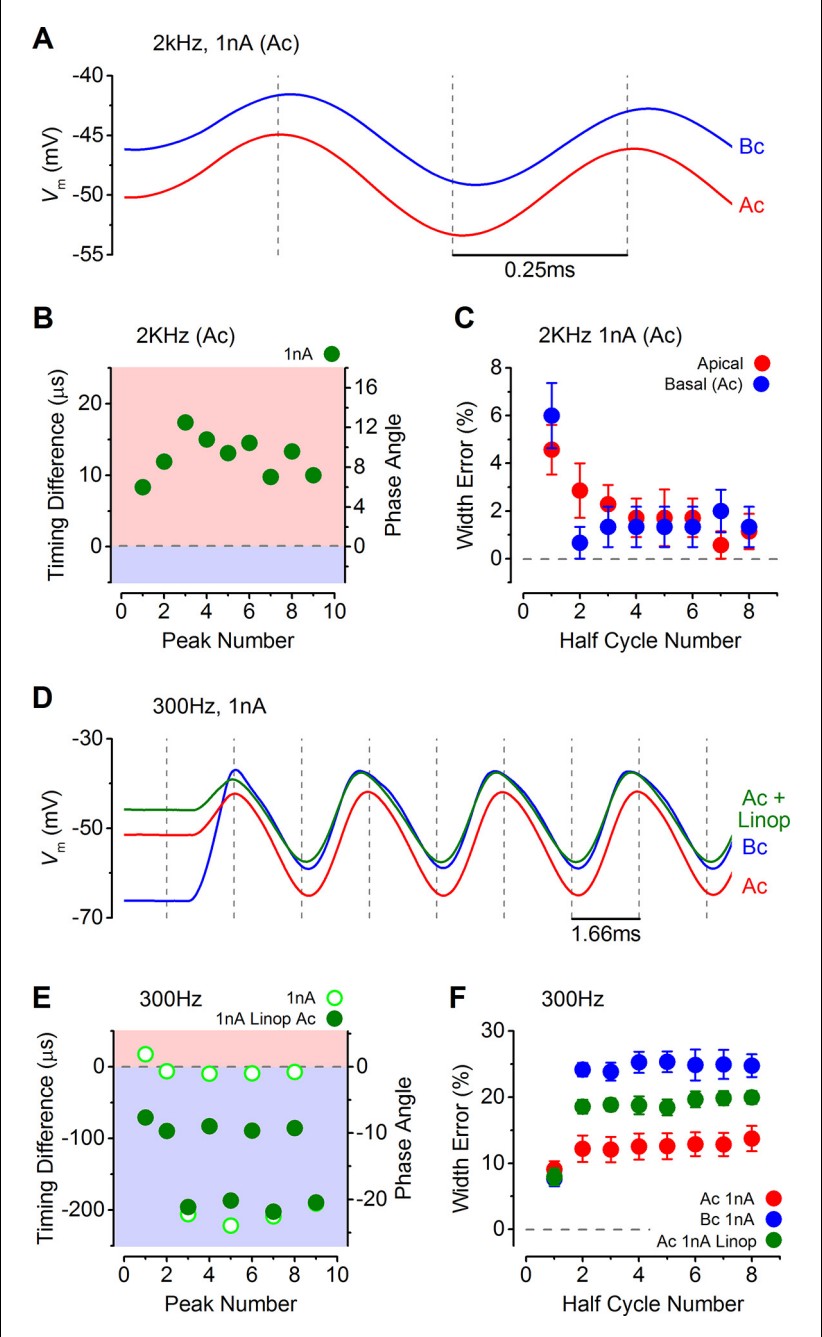

**Figure 9.** Response timing is determined by the intrinsic membrane properties of apical and basal inner hair cells (IHCs). (**A**) Average voltage responses of apical (n = 7) and basal (n = 6) IHCs to a 2 kHz 1 nA current sinewave using a stimulus with a holding current of 500 pA for both apical and basal IHCs. The dashed vertical lines are spaced to the 2 kHz half cycle and aligned to the initial apical peak. Even though both IHC types were stimulated with the same protocol, basal cells maintained a delayed response compared to apical cells. (**B**) The average timing difference between the arrival of the different peaks between apical and basal voltage responses shown in (**A**). Data points were obtained by subtracting the average apical value from that of basal cells (as in **Figure 8B**). All positive values indicate an apical lead. (**C**) Average half width errors, calculated as in **Figure 8C**, for the apical and basal voltage responses in A. Blocking $I_{K,n}$ in apical IHCs makes their response timing and accuracy more basal-like. (**D**) Average voltage responses of apical IHCs in the presence of 80 μM linopirdine (Ac+Linop, n = 6) to a 1 nA, 300 Hz current sinewave using a holding current of 500 pA for apical cells. Voltage responses in apical (Ac) and basal (Bc) IHCs are as in **Figure 7B** (holding current of 500 pA for apical and 50 pA for basal cells). The

*Figure 9. continued on next page*

*Figure 9. Continued*

dashed vertical lines are spaced to the 300 Hz half cycle and aligned to the initial apical peak (red trace). Note that when $I_{K,n}$ in apical cells was blocked by linopirdine their responses closely resembled those from basal cells with a similar leftward lean in the upper peaks following the initial one. Between –65 mV and –20 mV the $I_{K,n}$ represents a large component only in apical cells (*Figure 3D and E* and *Figure 4E*). (E) The average timing difference between the arrival of the different peaks between the voltage responses of apical and either basal or apical in linopirdine, shown in (D). (F) Average half width errors for the apical, apical in linopirdine and basal voltage responses in (D). The error values for apical cells in linopirdine were shifted towards those from basal values.

applied from the estimated in vivo resting $V_m$ (100 pA: *Figure 7A and C*). However, the more depolarized $V_m$ of apical IHCs, and the consequently larger resting membrane conductance, made positively going responses to stimulation generally smaller than in basal cells (*Figure 7A and B*). The hyperpolarizing phase of apical IHCs was more pronounced than in basal cells (*Figure 7B*). For larger stimuli, at frequencies approaching (2 kHz) or exceeding (10 kHz) the maximum limit for phase locking in vivo (*Palmer and Russell, 1986*), the first phase of the responses in apical IHCs reached its maximum value immediately after the onset of stimulation. By contrast, the responses of basal cells built up to reach a peak after a few cycles (*Figure 7D*) because they are superimposed on a rising DC response that grows with timing set by the membrane time constant. While apical IHCs appear to be suited for phasic signaling, basal cells are more adept at signaling graded responses to high-frequency stimuli (*Figure 7E and F*) since their receptor potential builds up to represent stimulus intensity.

The timing and accuracy of phasic responses in apical and basal IHCs were compared by measuring the time difference of the peaks and troughs of the first few cycles to sound-like stimulation (*Figure 8A*). The time differences are also shown as phase angle shifts, calculated based on the size of the time difference and the stimulus frequency. At 300 Hz the timing was generally more advanced in apical IHCs for the lower stimulus levels (100 pA: closed symbols in *Figure 8B*; see also *Figure 7A*). The same also occurred at 1 nA for the first peak, after which positive peaks in basal IHCs occurred sooner (*Figure 7B*), but this made the half-cycle width much less accurate in basal cells (*Figure 8C*). At 300 Hz the half-cycle error was generally lower at each stimulus level in apical IHCs. At 2 kHz, where the timing and accuracy of the phasic component would be vital in vivo, the peak responses of apical IHCs again occurred generally more rapidly than in basal cells (*Figure 8D*). In contrast to 300 Hz the timing difference at 2 kHz declined with progressive peaks to a stable level, but remained with an apical advance. This stable apical lead seems to be due to the underlying basolateral current differences between the two cell types (*Figures 3–5*), because when apical and basal cells were stimulated with the same apical-like 2 kHz 1 nA protocol a similar stable apical lead was evident (*Figure 9A–C*). Additionally, when the outward linopirdine-sensitive $K^+$ current, present only in apical cells (*Figure 4E*), was blocked, the voltage responses became more basal-like (*Figure 9D–F*). The 2 kHz responses of apical cells were very accurate at each stimulus level (*Figure 8E*), whereas basal cells began with a large half-cycle error that improved progressively, as seen for the timing differences. The decline in basal cell half-cycle error is likely to reflect the decrease in cell membrane time constant, to a fairly steady value, as depolarization activates the $K^+$ currents. Therefore, the intrinsic characteristics of apical IHCs seem to be best suited for the accurate representation of the phasic component of the receptor potential.

The maximal size of the MT current has been predicted to reach up to around 6 nA in vivo (*Kennedy et al., 2003*) due to the presence of a large endocochlear potential that drives ions through the MT channels, which is difficult to achieve in vitro. Therefore, a 6 nA sinewave stimulus (300 Hz) was applied to apical and basal IHCs (*Figure 10A and B*), in which the holding current was also increased to reflect the larger standing in vivo MT current. The responses of apical IHCs showed a slight depolarizing DC shift reaching a maximum of –25.7 ± 1.3 mV. Basal cells responded with larger changes in membrane potential, with very little or no hyperpolarization from rest and depolarizing up to –13.9 ± 1.6 mV, which corresponds to the peak of the calcium current (*Johnson and Marcotti, 2008*). The DC shift is likely to be slightly underestimated because a maximal MT current would show a greater saturation than that provided by the sinewave.

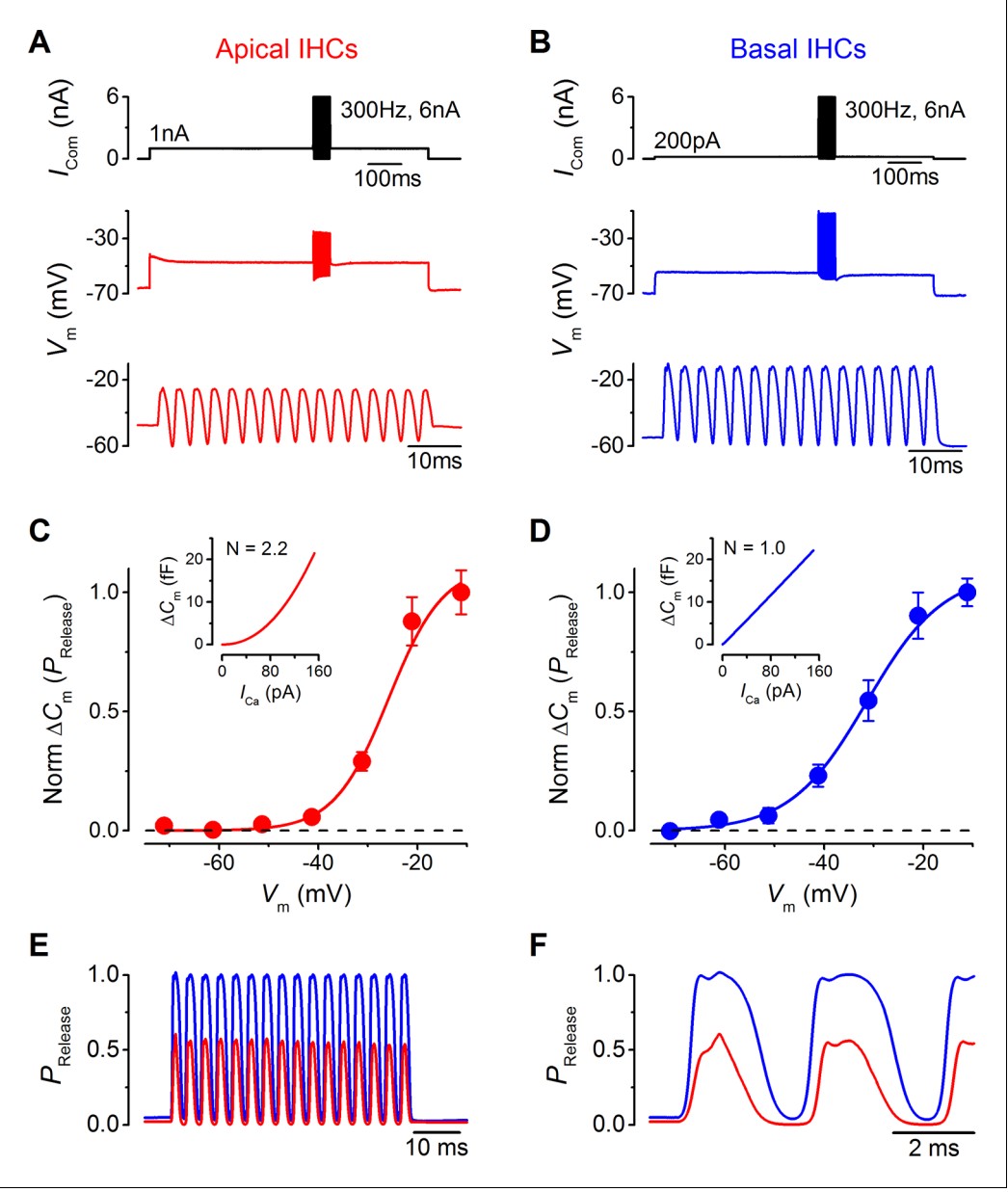

**Figure 10.** Maximal sound-like stimulation of apical and basal gerbil inner hair cells (IHCs) and predicted synaptic vesicle release probability suit phase-locking in apical cells. (**A** and **B**) Current clamp protocols (top panels) applied to apical (**A**) and basal (**B**) IHCs as in *Figure 6* but the holding currents and sinewave amplitudes have been increased to account for the contribution of the endocochlear potential (see Results). Average voltage responses from apical (n = 4) and basal (n = 3) IHCs are shown below with the sinewave region amplified in the lower panels. (**C** and **D**) Normalized average $\triangle C_m$ values plotted against membrane potential for apical (**A**) and basal (**B**) gerbil IHCs obtained from the synaptic transfer functions shown in the insets (from *Johnson et al., 2008*). The normalized $\triangle C_m$ approximates to a vesicle release probability ($P_{Release}$) and data points were fit with a Boltzmann curve with values of: apical IHCs $V_{half}$ –25.8 mV and $S$ 5.5 mV$^{-1}$; basal IHCs $V_{half}$ –31.4 mV and $S$ 7.7 mV$^{-1}$. (**E**) The average IHC voltage responses to maximal sound-like stimulation in (**A**) and (**B**) were transformed into $P_{Release}$ using the functions shown in (**C**) and (**D**) for apical and basal cells, respectively. (**F**) Same traces as in (**E**) but on an expanded time scale to emphasize the cycle-by-cycle differences in $P_{Release}$ between apical and basal cells, showing pronounced 'on' and 'off' responses to positive and negative phases of the sinusoidal currents, respectively.

In order to provide an understanding on how these maximal voltage responses would drive neurotransmitter release at IHC synapses, $V_m$ values were transformed into a probability of release ($P_{Release}$) using synaptic transfer functions previously obtained from adult apical and basal gerbil IHCs (*Johnson et al., 2008*). These synaptic transfer functions represent a relatively steady-state relation of IHC exocytosis (obtained using 100 ms voltage steps). I have used them to infer an overall $P_{Release}$ from all IHC synapses in this more dynamic situation based on the fact that IHC exocytosis is very fast (*Beutner et al., 2001*) and the release sites are within a nanodomain of the $Ca^{2+}$ channels (*Johnson et al., 2008*). However, there could be subtle differences in exocytosis kinetics for apical and basal IHCs especially amongst individual synapses that are likely to have different thresholds and operating ranges (*Ohlemiller et al., 1991*).

The high-order and linear exocytotic $Ca^{2+}$ dependence relations of apical (*Figure 10C*, inset) and basal cells (*Figure 10D*, inset) were used to express the normalized $\triangle C_m$ values as a function of cell $V_m$ (*Figure 10C and D*) and to obtain an approximated release probability relation. The fits to the data were used to transform the $V_m$ traces in *Figure 10A and B* into $P_{Release}$ (*Figure 10E and F*). $P_{Release}$ reached higher values in basal IHCs (0.98 ± 0.02, n = 3) than in apical cells (0.56 ± 0.07, n = 4). At the cell resting potential, $P_{Release}$ was not zero in either apical or basal IHCs (apical: 0.021 ± 0.002, n = 4; basal: 0.064 ± 0.022, n = 3; not significantly different), consistent with measured spontaneous activity in afferent fibers (*Ohlemiller et al., 1991*; *Ohlemiller and Siegel, 1994*). In apical IHCs $P_{Release}$ went below the resting level to reach almost zero during pronounced 'off' periods (responses to negative phases of the sinusoidal currents) that were longer lasting than those in basal cells where $P_{Release}$ did not reduce significantly below resting levels (apical minimal $P_{Release}$: 0.003 ± 0.001, n = 4; basal minimal $P_{Release}$: 0.038 ± 0.006, n = 3; $P<0.001$). The emphasized 'off' periods in apical IHCs (*Figure 10F*) would provide greater contrast for accurate phase-locking.

## Discussion

In this study I have shown that the biophysical characteristics of IHCs are specialized according to their frequency position within the gerbil cochlea. Using physiological $Ca^{2+}$ levels and body temperature, low-frequency (apical) IHCs had a significantly larger resting MT current than high-frequency (basal) cells. A large difference in the resting MT current was observed using three independent methods: direct MT current recordings, FM1-43 labeling, and shifts in the holding current in whole cell and perforated patch clamp. The physiological consequence of differences in the resting MT current is that the IHC resting membrane potential in vivo, in the absence of sound stimulation, is likely to be around –55 mV in apical and –65 mV in basal cells. This, combined with their different complement of basolateral membrane currents (*Figures 3–5*), indicates that low- frequency and high-frequency IHCs are intrinsically specialized to their location. The gerbil was the preferred model for this study because of its extended low-frequency hearing range (0.1–60 kHz: *Müller, 1996*) where phase-locking to auditory stimuli occurs (*Versteegh et al., 2011*), compared to the predominantly high-frequency hearing of the mouse (2–100 kHz: *Greenwood, 1990*).

### The biophysical properties of IHCs are specialized for the CF

The sound induced receptor potentials of low-frequency (below a few kilohertz) and high-frequency IHCs are very different. While low-frequency cells have a predominant sound frequency-following, or phase-locked, AC component (*Dallos, 1985*, *1986*; *Cheatham and Dallos, 1993*), the main component of high-frequency cells is a graded and sustained DC shift in potential in proportion to sound intensity (*Russell and Sellick, 1978*). Mammals use two main strategies to localize sounds of different frequency. Low frequencies are localized by principal cells in the medial superior olive (MSO) that compare the inter-aural timing differences (ITDs) of phase-locked activity in auditory afferents from the two ears (*Chirila et al., 2007*). High-frequency sounds are localized by principal cells in the lateral superior olive (LSO) that compares inter-aural level differences (ILDs) arising from the graded responses of IHCs of each ear (*Caird and Klinke, 1983*). Therefore, the biophysical properties of low-frequency and high-frequency IHCs should be best suited to maximize the accuracy of response-timing in apical cells and response-level in basal cells to allow the detection of ITDs down to 10 µs and ILDs of only 1–2 dB (*Grothe et al., 2010*). In the present study I found that low-frequency IHCs have a significantly more depolarized resting $V_m$ than basal cells arising from the large resting MT current, which increases their resting conductance and speeds up their voltage responses. In

response to simulated sound stimuli, apical IHCs were faster and preserved the phase width of the stimulus better than basal cells up to around the maximum frequency for phase-locking. However, the superior speed and timing of apical cells is not just down to their more depolarized $V_m$ since basal cells remained slower even when depolarized at similar values to those of apical cells (*Figure 9A–C*). I found that the differential expression of the $K^+$ current $I_{K,n}$ in apical and basal IHCs was a major determinant for response speed and timing at low frequencies (*Figure 9D–F*).

The depolarized resting $V_m$ of apical cells (about –55 mV) caused their 'off' response to sound-like stimulation to be larger than their 'on' component (responses to positive phases of the sinusoidal currents) for moderate stimuli (*Figure 5B*). This is comparable to the visual system where photons of light cause a hyperpolarization of the photoreceptor membrane (*Baylor et al., 1979*) and a reduction in the release of glutamate. This 'backwards' mechanism introduces much less noise into the signal and enhances contrast, which would be favorable for phase-locking to auditory stimuli (*Palmer and Russell, 1986*). The high-order $Ca^{2+}$ dependence of synaptic vesicle release in apical gerbil IHCs (*Johnson et al., 2008*) enhances these 'off' periods when $V_m$ is transformed into $P_{Release}$ (*Figure 10*), which would further increase the signal contrast as it is relayed onto afferent fibers and push the boundary for phase-locking up to a higher frequency. The smaller overall $P_{Release}$ of apical IHCs to the maximal predicted stimulation would be compensated by multivesicular (*Glowatzki and Fuchs, 2002*) or uniquantal (*Chapochnikov et al., 2014*) release at individual ribbon synapses, generating large excitatory postsynaptic potentials (EPSPs) in the afferent terminals, the size of which has been shown to be $Ca^{2+}$ independent and larger ones show more accurate phase-locking (*Glowatzki and Fuchs, 2002*; *Grant and Glowatzki, 2010*; *Li et al., 2014*). Recent findings have shown that a more depolarized membrane potential in hair cells leads to the facilitation of glutamate release (*Cho et al., 2011*; *Goutman and Glowatzki, 2011*), which would have important consequences for IHC neurotransmitter release and short-term plasticity, and spike adaptation during a sound stimulus. However, a generally smaller exocytotic response, as calculated for the maximal $P_{Release}$ in apical IHCs (*Figure 10E and F*), would again be favorable for maximizing response speed, at the expense of accurately signaling intensity. This could explain why sound intensity discrimination is better towards the higher frequencies (1–6 kHz; *Fletcher and Munson, 1933*).

While apical cells are adapted for response speed, basal cells are specialized for intensity coding. Their more hyperpolarized $V_m$ (about –65 mV), due to the much smaller resting MT current, and consequently greater membrane time constant, means that the responses of basal cells to even low level sounds summate into a DC membrane potential shift that is proportional to stimulus intensity. By contrast, apical IHCs only showed signs of a DC component for large stimuli (*Figure 10A*). The lower resting conductance of basal cells also means that their response amplitudes are bigger, although slower, than apical cells and are more spread out giving better discrimination between stimulus levels (*Figure 2E*). The overall linear $Ca^{2+}$ dependence of neurotransmitter release in adult gerbil basal IHCs (*Johnson et al., 2008*) would extend the dynamic range, enabling cells to accurately relay information from both low and high intensity sounds into afferent activity.

The differences in mature gerbil IHCs seen here are opposite to what we previously observed in mature OHCs (*Johnson et al., 2011*). In OHCs it is the high-frequency basal cells that have the fastest membrane properties and largest currents. This is important because they follow sound up to the highest frequencies in order to drive the prestin motors on a cycle-by-cycle basis for cochlear amplification (*Dallos et al., 2006*; *Fettiplace and Hackney, 2006*; *Ashmore, 2008*). In order for IHCs to be fast enough to encode the phasic component of high-frequency sounds, the size of the receptor potential would be so small that the graded component would be lost and the response would be too fast to be encoded by the release of synaptic vesicles.

## IHCs play a crucial role in auditory information processing

Classical in vivo sharp electrode recordings of IHC receptor potential responses to sound reveal similar distinctions between apical and basal IHCs (*Palmer and Russell, 1986*; *Cheatham and Dallos, 1993*). Low-frequency guinea pig IHCs respond to sound with a predominant AC component and a large hyperpolarization from rest (*Dallos, 1986*). By contrast, the responses in high-frequency cells are dominated by the DC component (*Russell and Sellick, 1978*), although when stimulated with low-frequency sound they also show an AC component but with little hyperpolarization from rest (*Palmer and Russell, 1986*), indicative of a smaller resting MT current in basal cells. In vivo studies on IHCs and afferent fibers have independently suggested that there are intrinsic differences

between apical and basal regions, which generate a resting bias for apical cells towards the 'on' or depolarized condition and basal cells to the 'off' hyperpolarized condition (*Palmer and Russell, 1986*; *Ohlemiller and Siegel, 1994*; *Cheatham and Dallos, 1993*). A different resting bias would result from differences in the standing current through the IHCs (i.e., resting MT current), such that the larger current in apical cells would reduce the DC component to favor AC responses, whereas the smaller current in basal cells would enhance the DC $V_m$ shifts. A different resting bias would also explain why afferent fibers in the gerbil generally have a higher spontaneous firing rate (SR) at the apex than those at base (*Ohlemiller et al., 1991*; *Ohlemiller and Siegel, 1994*). Therefore, both low-frequency and high-frequency IHCs are specialized for their respective tasks, showing a preference for either timing or intensity coding, respectively. While low-frequency cells are able to respond rapidly and accurately to phasic stimuli at around the phase-locking limit, high-frequency cells show clearly defined graded shifts in membrane potential over an extended dynamic range. Evidence in support of a specialization of apical cells for phase-locking comes from in vivo recordings from chinchilla auditory nerve fibers showing that the strength of phase-locking in high-CF fibers starts to deteriorate at lower stimulus frequencies than that in low-CF fibers (*Temchin and Ruggero, 2010*). The findings presented here show that the primary sensory IHCs play a more active role in auditory information processing than previously thought.

## Materials and methods

### Electrophysiology

Gerbil cochlear IHCs ($n$ = 115) were studied in acutely dissected organs of Corti from postnatal day 17 (P17) to P40, where the day of birth is P0. Animals of either sex were killed by cervical dislocation, under schedule 1 in accordance with UK Home Office regulations. IHCs were positioned at a frequency range of 250–420 Hz in apical and 20–37 kHz in basal cells (*Müller, 1996*). Cochleae were dissected as previously described (*Johnson et al., 2008*, *2012*) in normal extracellular solution (in mM): 135 NaCl, 5.8 KCl, 1.3 CaCl$_2$, 0.9 MgCl$_2$, 0.7 NaH$_2$PO$_4$, 5.6 D-glucose, 10 Hepes-NaOH. Sodium pyruvate (2 mM), MEM amino acids solution (50X, without L-Glutamine), and MEM vitamins solution (100X) were added from concentrates (Fisher Scientific, UK). The pH was adjusted to 7.5 (osmolality ~308 mmol/kg). The dissected organs of Corti were transferred to a microscope chamber, immobilized using a nylon mesh fixed to a stainless steel ring, and continuously perfused with the above extracellular solution. The organs of Corti were observed with an upright microscope (Nikon, Japan) with Nomarski differential interference contrast optics (X60 water immersion objective and X15 eyepieces).

Whole-cell patch clamp recordings were performed at body temperature (34–37°C) using an Optopatch (Cairn Research Ltd, UK) amplifier. Patch pipettes (2–3 MΩ) were coated with surf wax (Mr. Zogs SexWax, USA) to minimize the fast capacitance transient of the patch pipette. The pipette intracellular solution contained (in mM): 131 KCl, 3 MgCl$_2$, 1 EGTA-KOH, 5 Na$_2$ATP, 5 Hepes-CsOH, 10 Na$_2$-phosphocreatine (pH 7.3; osmolality ~296 mmol/kg).

The perforated-patch technique (*Rae et al., 1991*) was used on a few mature gerbil IHCs ($n$ = 7) in order to see whether endogenous Ca$^{2+}$ buffering affected the resting MT current and the other basolateral membrane currents. For these experiments the pipette filling solution contained (mM): 21 KCl, 110 potassium aspartate, 3 MgCl2, 5 Na2ATP, 1 EGTA–KOH, 5 Hepes–KOH, 10 sodium phosphocreatine (pH 7.3, 295 mmol/kg). The antibiotic amphotericin B (Calbiochem, UK) was dissolved in dry dimethylsulfoxide (DMSO) prior to its dilution into the above intracellular solution to a final concentration of 120–240 µg/ml. The patch pipette was tip-filled with the above normal potassium aspartate intracellular solution before back-filling with the amphotericin B containing solution to prevent leakage of the antibiotic onto the IHC prior to sealing onto the membrane. Data acquisition was controlled by pClamp software using a Digidata 1440A (Molecular Devices, USA). Voltage-clamp recordings were low-pass filtered at 2.5 kHz (8-pole Bessel) and sampled at 5 kHz. Current clamp steps were recorded at 5 kHz and filtered at 2.5 kHz and the sound-like stimulation protocols were recorded at 100 kHz and filtered at 20 or 50 kHz. Data analysis was performed using Origin software (OriginLab, USA). The residual series resistance ($R_s$) after compensation (60–80%) was 1.16 ± 0.08 MΩ ($n$ = 60) for apical IHCs and 1.15 ± 0.09 MΩ ($n$ = 55) for basal cells. The average voltage-clamp time constant (product of $R_s$ and membrane capacitance $C_m$; apical: 12.4 ± 0.1 pF, $n$ = 60,

basal: 11.7 ± 0.2 pF, $n$ = 55) was 14.5 ± 1.0 µs for apical IHCs and 13.2 ± 0.8 µs for basal cells. Membrane potentials were corrected for a liquid junction potential measured between electrode and bath solutions of –4 mV for the KCl intracellular solution or –10 mV for the K-aspartate solution.

In some voltage and current clamp experiments an extracellular solution containing low-$Ca^{2+}$ (40 µM $Ca^{2+}$: obtained by buffering 3.7 mM $Ca^{2+}$ with 4 mM (2-hydroxyethyl)ethylenediaminetriacetic acid) was used to mimic the endolymphic $Ca^{2+}$ concentration (20–40 µM: *Bosher and Warren, 1978*; *Salt et al., 1989*). To verify that the effects of the low-$Ca^{2+}$ solution were mediated by the transducer channel, the transducer channel blocker dihydrostreptomycin (DHS; 100 µM, *Marcotti et al., 2005*) was added to the low-$Ca^{2+}$ solution. In order to isolate and identify the different $K^+$ currents expressed in IHCs ($I_{K,s}$, $I_{K,n}$, $I_{K,f}$: *Marcotti et al., 2003*, *2004*), different ion channel blockers were added to the extracellular solution: the KCNQ channel blocker linopridine (80 µM), the BK $Ca^{2+}$-activated $K^+$ channel blocker iberiotoxin (60 nM), both from Tocris Bioscience (UK), and the inward rectifier channel blocker $Cs^+$ (5 mM). Extracellular solutions containing different ion concentrations or channel blockers were topically perfused onto IHCs using a multi-barreled pipette positioned close to the cells, but far enough away as to not cause any hair bundle movement.

Statistical comparisons of means were made by Student's two-tailed $t$-test or, for multiple comparisons, analysis of variance, usually one-way ANOVA followed by the Tukey test. Mean values are quoted ± s.e.m. where $P<0.05$ indicates statistical significance.

## Hair bundle stimulation

MT currents were elicited by stimulating the hair bundles of apical and basal coil mature gerbil cochlear IHCs (P17–P25) using a fluid jet from a pipette (tip diameter 10–12 µm) driven by a piezoelectric disc (*Corns et al., 2014*, *Kros et al., 1992*). The pipette tip of the fluid jet was positioned near to the bundles to elicit a maximal transducer current. Mechanical stimuli were applied as 50 Hz sinusoids (filtered at 0.5 kHz, 8-pole Bessel) with driving voltages of up to ± 20 V. Positive driver voltage (fluid flowing out of the jet) caused hair bundles to move towards the taller stereocilia and produced excitatory responses. The pressure in the fluid jet was zeroed before approaching the IHC to ensure hair bundles were not displaced from their resting position. All recordings were carried out at body temperature and in the presence of the endolymph-like low-$Ca^{2+}$ solution (40 µM).

## FM1-43 labeling

FM1-43 experiments were performed on acutely dissected cochleae as previously described (*Gale et al., 2001*). Briefly, the apical and basal cochlear coils from the mature gerbil (P18–P22) were dissected in normal extracellular solution and then placed next to each other in the recording chamber and held down using a nylon mesh. At this stage the normal extracellular solution containing 1.3 mM $Ca^{2+}$ was changed to the low-$Ca^{2+}$ (40 µM) solution. The cochlear coils were then bath exposed to a low-$Ca^{2+}$ solution containing 3 µM FM1-43 for 8 s and immediately washed several times with normal 1.3 mM $Ca^{2+}$ extracellular solution. The cochleae were then viewed with an upright microscope (Olympus, Japan) equipped with epifluorescence optics and fluorescein isothiocyanate (FITC) filters (excitation, 488 nm; emission, 520 nm) using a X63 water immersion objective. Images were captured using a charge-coupled device (CCD) camera and were taken within 15 min after exposure to FM1-43. Fluorescence images were taken with 4 s exposure time. Stock solutions of 3 mM FM1-43 were prepared in water. A total number of six cochleae from three gerbils were used and the tectorial membranes were left intact. These experiments were performed at room temperature (22–25°C).

## Acknowledgements

This work was supported by grants from The Royal Society to SLJ. SLJ is a Royal Society University Research Fellow. I thank W Marcotti, MC Holley and CJ Kros for commenting on the manuscript and/or advice on the analysis of the data.

## Additional information

### Funding

| Funder | Grant reference number | Author |
|---|---|---|
| Royal Society | University Research Fellowship | Stuart L Johnson |

The funders had no role in study design, data collection and interpretation, or the decision to submit the work for publication.

### Author contributions

SLJ, Conception and design, Acquisition of data, Analysis and interpretation of data, Drafting or revising the article

### Ethics

Animal experimentation: Animals were killed by cervical dislocation, under schedule 1 in accordance with UK Home Office regulations. All animal studies were licensed by the U.K. Home Office under the Animals (Scientific Procedures) Act 1986 and were approved by the University of Sheffield Ethical Review Committee.

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
