## [Decision Letter]

Thank you for submitting your work entitled "Membrane properties specialise mammalian inner hair cells for frequency or intensity encoding" for peer review at *eLife*. Your submission has been favorably evaluated by Eve Marder (Senior editor), a Reviewing editor (Andrew King), and three reviewers, two of whom have agreed to reveal their identity: Ruth Anne Eatock and Henrique von Gersdorff.

The reviewers have discussed the reviews with one another and the Reviewing editor has drafted this decision to help you prepare a revised submission.

Summary:

Your paper has been considered by three reviewers, who agree that your study is technically proficient and that the differences you report in the transduction currents and input resistance between apical and basal inner hair cells potentially have important functional implications for sensory coding. In particular, they noted that characterizing ionic currents in isolated cochleas of adult gerbils under conditions comparable to those in vivo is a significant accomplishment. Concern was expressed, however, that the observed differences in the probability of opening of the transducer channel at rest might be due to the recording conditions used. Consequently, this and several other issues will need to be addressed before a final decision can be made.

1) The possibility that the difference in channel open probabilities could be attributable to differences in intracellular calcium buffering, as seen in other hair cells, needs to be addressed. The reviewers discussed the value of using perforated-patch recordings, where the cytoplasmic proteins are not washed out, or of using various calcium buffer concentrations in conjunction with whole-cell recording to address this issue. The same point was raised in the context of measurements of resting potential and membrane time constants. While they agreed that perforated-patch recordings present their own difficulties and are not essential for this study, the possibility that differences in calcium loading between basal and apical inner hair cells have to be considered. One of the reviewers also noted that there is a prior report of a tonotopic gradient in calcium-binding proteins (Pack and Slepecky, 1995).

2) The reviewers stated that the use of current injection may not have been the best approach for characterization of the difference in sound-evoked receptor potentials, since this is not accompanied by the related conductance change, and whether direct stimulation of the hair bundle or dynamic clamp would have been better. You will need to address this issue in any revision.

3) All the reviewers thought that the FM1-43 results are important and need to be illustrated in the manuscript.

4) One of the reviewers questioned the use of frequency domain (sines) to study signaling and then analysis of the results exclusively and seemingly laboriously in the time domain. You should consider whether the effects of various cellular processes on timing and linearity of the response could be efficiently measured as phase angles and the presence of harmonics in the receptor potential. These numbers could then be compared to analogous numbers from in vivo responses of auditory afferents.

5) Please consider your findings in relation to the phase-locking of auditory nerve fibres: does previous evidence show that if you drive a basal high-CF cochlear fibre with a loud, low-frequency tone, its phase-locking is worse than that of a low-CF fiber at low frequencies?

[Editors' note: further revisions were requested prior to acceptance, as described below.]

Thank you for resubmitting your work entitled "Membrane properties specialize mammalian inner hair cells for frequency or intensity encoding" for further consideration at *eLife*. Your revised article has been favorably evaluated by Eve Marder (Senior editor), a Reviewing editor, and two reviewers.

The reviewers agree that the addition of the perforated patch experiments has strengthened the manuscript and that the technical issues previously raised have been dealt with satisfactorily. However, there are some remaining issues that need to be addressed before acceptance, as outlined below. In particular, the reviewers found the Results section hard to follow in places and have made the following specific comments:

1) Your global replacement of "tuning" with "specialization", in response to a reviewer's suggestion, works in some contexts, but, in other cases, "filtering" would do the job better – it gets across the idea of frequency (which specialization does not) without implying sharp tuning.

2) Your response to the suggestion to use dynamic clamp has been questioned: It may not be easy to implement dynamic clamp at these frequencies, and is not certain to substantially affect the data.

3) In paragraph two of the Introduction, do you mean "too fast to fully charge the cell membrane" or "too fast for the cell membrane potential to keep up"?

4) In paragraph two of the subsection “Tonotopic differences in IHC basolateral membrane currents,” the statement that DHS prevented the increase in holding current (i.e. depolarization of resting potential) is true for the apical cells, but is not shown to be true in the basal cell exemplar (Figure 5). Are there other examples for which adding DHS did a good job of blocking the "leak" conductance at –64 mV (as is not the case in Figure 5)? Otherwise you may need to back off on that assertion for basal cells.

5) Paragraph four of the subsection “Voltage responses of IHCs to sound-like stimulation” and Figure 8. The large "error" at the start of the tone burst stimulus presumably reflects the rapid decrease in tau-m as depolarization activates K currents – affecting the timing of the response even in the course of one stimulus cycle. The cycle width accuracy improves once tau-m has a new fairly steady value, so that the same phase lag (delay) holds throughout the cycle. The text currently does not make this clear.

6) Paragraph five, subheading “Voltage responses of IHCs to sound-like stimulation”. Don't you mean the larger "standing" in vivo MT current – rather than "asymmetric"? (A steady current can never be symmetric, at least in the way "symmetric" has been used in this paper, so the use of the term "asymmetric" is puzzling).

7) In paragraph seven of the same subsection, indicate what the Prelease levels at resting potential are, perhaps on Figure 10. The troughs of the Prelease plots are both so close to zero, it is hard to see that there is a significant difference between apical and basal cells in this regard.

8) In paragraph one of the Discussion, a reference is given as a number (29).

9) Paragraph two, subsection “The biophysical properties of IHCs are specialized for the characteristic frequency”: Sometimes the writing is too telegraphic for me to follow. Are you again commenting on the observation that as frequency increases, there is little DC component to drive release in the apical hair cells, in contrast to the basal cells?

10) Paragraph three, subsection “The biophysical properties of IHCs are specialized for the characteristic frequency”. When you say that the more hyperpolarised *V*_m_ acts to create a DC shift, do you mean that the hyperpolarised *V*_m_ does this by keeping ion channels closed and thereby increasing tau-m or is any additional mechanism implied here?

11) I'm not sure "on-like" and "off-like" will be clear to the reader. How about just stating somewhere in a legend or in the text that by "on" and "off" you mean responses to positive and negative phases of the sinusoidal currents, respectively?

---

## [Author Response]

*[…] Consequently, this and several other issues will need to be addressed before a final decision can be made.*

*1) The possibility that the difference in channel open probabilities could be attributable to differences in intracellular calcium buffering, as seen in other hair cells, needs to be addressed. The reviewers discussed the value of using perforated-patch recordings, where the cytoplasmic proteins are not washed out, or of using various calcium buffer concentrations in conjunction with whole-cell recording to address this issue. The same point was raised in the context of measurements of resting potential and membrane time constants. While they agreed that perforated-patch recordings present their own difficulties and are not essential for this study, the possibility that differences in calcium loading between basal and apical inner hair cells have to be considered. One of the reviewers also noted that there is a prior report of a tonotopic gradient in calcium-binding proteins (Pack & Slepecky, 1995).*

I agree that possible differences in the buffering capability could affect the main conclusions. However, published data from IHCs of the adult gerbil cochlea has demonstrated that the buffering strength is likely to be similar between apical and basal cells (Johnson et al., 2008). Despite this, I decided to verify this previous finding by recording voltage responses from mature apical and basal gerbil IHCs (P29-P36) using new perforated patch experiments. Rather than performing MT current recordings on mature IHCs using the fluid jet, which are extremely complex experiments even in whole-cell configuration, I used the cells holding current at -64 mV as a measure of the resting transducer current, which is responsible for determining the level of depolarization of the IHC at rest. The new data show that the holding current measured in endolymphatic extracellular Ca^2+^ was much larger in apical cells than basal cells. This holding current was linked to the MT current because it was selectively abolished by the application of the MT channel blocker DHS, indicating that the results in whole-cell are a true representation of the physiological condition in mature IHCs. This also shows that, as previously demonstrated (Johnson et al., 2008), the observed differences between apical and basal IHCs are not a direct consequence of possible differences in the endogenous Ca^2+^ buffer along the cochlea. Of course, this is unlikely to apply to OHCs (Johnson et al., 2011, Neuron). The new data are shown in the new Figure 5, described in the Results section (paragraphs two and three, subsection “Tonotopic differences in IHC basolateral membrane currents”) and detailed in the second paragraph of the Methods.

*2) The reviewers stated that the use of current injection may not have been the best approach for characterization of the difference in sound-evoked receptor potentials, since this is not accompanied by the related conductance change, and whether direct stimulation of the hair bundle or dynamic clamp would have been better. You will need to address this issue in any revision.*

I believe that current injection was the best approach to characterise the difference in sound evoked potentials since the protocols were designed to mimic the MT current. Direct stimulation of the hair bundle would not be appropriate for this experiment mainly because the tall hair bundle of adult IHCs are easily damaged by repeated stimulation, which was a requirement for to collecting accurate and reliable data. Hair bundle damage would lead to differences in the stimulus intensity within the same cell and from cell to cell, making it very difficult to compare apical and basal cells. Dynamic clamp would do the same thing as injecting current in this case since it predicts the resulting current through the MT channels and applies it as a current injection. Dynamic clamp uses the conductance values to work out the current through the channel at different potentials, so effectively adds the channel to the cell. However, I just want to mimic the transducer current from the resting potential of the cell, so if I know what it should be at rest and during stimulation I can apply it to the cell and see how it responds. So, in some respect I am already doing dynamic clamp.

*3) All the reviewers thought that the FM1-43 results are important and need to be illustrated in the manuscript.*

The FM1-43 results have been added to Figure 1.

*4) One of the reviewers questioned the use of frequency domain (sines) to study signaling and then analysis of the results exclusively and seemingly laboriously in the time domain. You should consider whether the effects of various cellular processes on timing and linearity of the response could be efficiently measured as phase angles and the presence of harmonics in the receptor potential. These numbers could then be compared to analogous numbers from in vivo responses of auditory afferents.*

As suggested, I have provided the phase angles in degrees for the timing differences as well so readers can make a comparison with the auditory nerve literature. These are shown in Figure 8 and Figure 9 where I have included an additional y-axis on the graphs showing phase angles. Regarding harmonics, I haven't done a Fourier analysis of the data because the non-linearities that I report are strongly time-dependent (as can be seen from panels B-E in the new Figure 8). Fourier analysis provides a plot of frequency components of recording, but works only with steady-state responses.

5) Please consider your findings in relation to the phase-locking of auditory nerve fibres: does previous evidence show that if you drive a basal high-CF cochlear fibre with a loud, low-frequency tone, its phase-locking is worse than that of a low-CF fiber at low frequencies?

This is a very interesting remark. A recent study by Temchin and Ruggero in JARO (2010) has mentioned that phase-locking of high-CF fibres starts to deteriorate at lower stimulus frequencies than that of low-CF fibres. This paper is now mentioned in the Discussion (in the subsection “IHCs play a crucial role in auditory information processing”).

[Editors' note: further revisions were requested prior to acceptance, as described below.]

*The reviewers agree that the addition of the perforated patch experiments has strengthened the manuscript and that the technical issues previously raised have been dealt with satisfactorily. However, there are some remaining issues that need to be addressed before acceptance, as outlined below. In particular, the reviewers found the Results section hard to follow in places and have made the following specific comments:*

*1) Your global replacement of "tuning" with "specialization", in response to a reviewer's suggestion, works in some contexts, but, in other cases, "filtering" would do the job better – it gets across the idea of frequency (which specialization does not) without implying sharp tuning.*

I have changed “specialization” to “filtering” in two places (at the end of the Abstract and Introduction).

*2) Your response to the suggestion to use dynamic clamp has been questioned: It may not be easy to implement dynamic clamp at these frequencies, and is not certain to substantially affect the data.*

I agree with the comment and this is the reason why I used current injection to mimic the transducer current.

*3) In paragraph two of the Introduction, do you mean "too fast to fully charge the cell membrane" or "too fast for the cell membrane potential to keep up"?*

As suggested, I have added “potential” to the sentence.

*4) In paragraph two of the subsection “Tonotopic differences in IHC basolateral membrane currents,” the statement that DHS prevented the increase in holding current (i.e. depolarization of resting potential) is true for the apical cells, but is not shown to be true in the basal cell exemplar (Figure 5). Are there other examples for which adding DHS did a good job of blocking the "leak" conductance at*

*–64 mV (as is not the case in Figure 5)? Otherwise you may need to back off on that assertion for basal cells.*

I do not agree with this comment since the use of DHS is only aimed at blocking the transducer channel component of the holding current (not “leak” current). The reason why in basal IHCs there is a larger “leak” compared to control is because DHS is present together with Low Ca, which per se will reduce a BK current (Marcotti et al., 2004). Indeed, DHS does reduce the holding current in basal cells, although to a lesser extent to apical IHCs due to a much smaller resting transducer current. However, we have now modified the text to clarify that the reduced holding current in DHS refers to the transducer current component (paragraph two, subsection “Tonotopic differences in IHC basolateral membrane currents”).

*5) Paragraph four of the subsection “Voltage responses of IHCs to sound-like stimulation” and Figure 8. The large "error" at the start of the tone burst stimulus presumably reflects the rapid decrease in tau-m as depolarization activates K currents – affecting the timing of the response even in the course of one stimulus cycle. The cycle width accuracy improves once tau-m has a new fairly steady value, so that the same phase lag (delay) holds throughout the cycle. The text currently does not make this clear.*

This is a good point and I have now added a sentence to explain this more clearly.

*6) Paragraph five, subheading “Voltage responses of IHCs to sound-like stimulation”. Don't you mean the larger "standing" in vivo MT current – rather than "asymmetric"? (A steady current can never be symmetric, at least in the way "symmetric" has been used in this paper, so the use of the term "asymmetric" is puzzling).*

I changed “asymmetric” to “standing”.

*7) In paragraph seven of the same subsection, indicate what the Prelease levels at resting potential are, perhaps on Figure 10. The troughs of the Prelease plots are both so close to zero, it is hard to see that there is a significant difference between apical and basal cells in this regard.*

I added the values for Prelease at rest and during the troughs (paragraph seven, same subsection).

*8) In paragraph one of the Discussion, a reference is given as a number (29).*

Thank you for spotting this oversight, which I have amended (the reference is to Muller, 1996).

*9) Paragraph two, subsection “The biophysical properties of IHCs are specialized for the characteristic frequency”: Sometimes the writing is too telegraphic for me to follow. Are you again commenting on the observation that as frequency increases, there is little DC component to drive release in the apical hair cells, in contrast to the basal cells?*

My intension was to highlight the fact that the change in Prelease in apical cells doesn’t reach the same maximal values as those in basal cells, which is due to the smaller *V*_m_ changes combined with the high-order Ca dependent exocytosis. I now mention in the text that this refers to the smaller maximal Prelease calculated for apical IHCs.

*10) Paragraph three, subsection “The biophysical properties of IHCs are specialized for the characteristic frequency”. When you say that the more hyperpolarised* V*_m_ acts to create a DC shift, do you mean that the hyperpolarised* V*_m_ does this by keeping ion channels closed and thereby increasing tau-m or is any additional mechanism implied here?*

The DC shift for even low levels of stimulation is mainly due to the much smaller resting MT current in basal cells, which also contributes to the larger tau-m. Therefore once the MT current is larger than 100pA (twice the resting MT current) there will be a DC shift in *V*_m_. I now mention it is due to the small resting MT current in the text.

*11) I'm not sure "on-like" and "off-like" will be clear to the reader. How about just stating somewhere in a legend or in the text that by "on" and "off" you mean responses to positive and negative phases of the sinusoidal currents, respectively?*

I have defined “off” and “on” in the text (paragraph seven, subsection “Voltage responses of IHCs to sound-like stimulation” and paragraph two, subsection “The biophysical properties of IHCs are specialized for the characteristic frequency”) and in the legend of Figure 10.